# Object-Centric Relational Representations for Image Generation

**Luca Butera**                                                    *luca.butera@usi.ch*
*The Swiss AI Lab IDSIA & Università della Svizzera italiana*

**Andrea Cini**                                                    *andrea.cini@usi.ch*
*The Swiss AI Lab IDSIA & Università della Svizzera italiana*

**Alberto Ferrante**                                              *alberto.ferrante@usi.ch*
*The Swiss AI Lab IDSIA & Università della Svizzera italiana*

**Cesare Alippi**                                                  *cesare.alippi@usi.ch*
*The Swiss AI Lab IDSIA & Università della Svizzera italiana*
*Politecnico di Milano*

**Reviewed on OpenReview:** *https://openreview.net/forum?id=7kWjB9zW9O*

## Abstract

Conditioning image generation on specific features of the desired output is a key ingredient of modern generative models. However, existing approaches lack a general and unified way of representing structural and semantic conditioning at diverse granularity levels. This paper explores a novel method to condition image generation, based on object-centric relational representations. In particular, we propose a methodology to condition the generation of objects in an image on the attributed graph representing their structure and the associated semantic information. We show that such architectural biases entail properties that facilitate the manipulation and conditioning of the generative process and allow for regularizing the training procedure. The proposed conditioning framework is implemented by means of a neural network that learns to generate a 2D, multi-channel, layout mask of the objects, which can be used as a soft inductive bias in the downstream generative task. To do so, we leverage both 2D and graph convolutional operators. We also propose a novel benchmark for image generation consisting of a synthetic dataset of images paired with their relational representation. Empirical results show that the proposed approach compares favorably against relevant baselines.

## 1 Introduction

Graphs are an abstraction that allows for representing objects as collections of entities and binary relationships. Previous research on graph-based image generation has been limited to the high-level conditioning of the image content by means of *scene graphs*, i.e., graphs where nodes represent objects, and edges denote subject-predicate-object relationships (Johnson et al., 2018; Ivgi et al., 2021). Conversely, conditioning on desired fine-grained properties, e.g., spatial location, arrangement or visual attributes, is usually performed without considering a relational structure. In fact, most of the literature dealing with pose-constrained image generation, e.g., (Ma et al., 2017; Siarohin et al., 2018; Neverova et al., 2018; Qian et al., 2018; Jiang et al., 2022), represents key points and semantic attributes with 2D masks or feature vectors; hence not exploiting known relationships among the components of the object being generated. Indeed such approaches are usually limited to conditioning the generation on a fixed template structure and lack flexibility. This paper aims at taking full advantage of graph representations in this context; our approach fills the above research gap and addresses the shortcomings of existing methods by encoding relational inductive biases into

the processing. In our method, graphs are used both as effective and compact, fine-grained object-centric representations of the image content as well as an architectural bias for the neural architecture. We call such representations *attributed pose graphs*, i.e., graphs whose nodes can represent particular landmarks in an object's structure and have both positional (location) and, possibly, semantic attributes (e.g., color, class). Edges, instead, account for relationships among nodes, capturing the object's morphology. Our formulation allows for encoding all the desired properties of the generated image in such a graph, without relying on additional inputs, such as reference images or similar, allowing for high flexibility in specifying the conditioning. The generation of the scene can then be manipulated, without any change to the model architecture, by editing the graph, e.g., by modifying the nodes' attributes and/or the number of landmarks used to represent each object. This is thanks to the exploitation of *neural message passing* (Gilmer et al., 2017) and *graph neural networks* (GNNs) (Bacciu et al., 2020; Bronstein et al., 2021) used for constraining the processing.

Our model, named *GraPhOSE*, relies on learning a multi-channel layout mask from the structured representation of the scene. It is trained jointly with a downstream model (image decoder) that contextually exploits such mask to constrain the generative process. To overcome the lack of pre-annotated object masks for specific use cases, we also propose the usage of *surrogate masks* corresponding to procedurally generated synthetic pose graphs, as targets for a supervised pre-training of GraPhOSE. After the pertaining, the full architecture, i.e., GraPhOSE and the downstream model, can be trained end-to-end on the task at hand. Additionally, since pre-training relies on randomly generated graphs potentially (partially) outside the target distribution, the proposed method can act as a regularization, preventing overfitting the most common poses in the target dataset.

To the best of our knowledge, this is the first work to use object-centric graph-based relational representations to condition a neural generative model. Furthermore, we complement the methodological contributions by introducing a benchmark for pose-conditioned image generation: we propose a synthetic dataset named *Pose-Representable Objects* (PRO), which aims at assessing the performance of generative models in conditioning image generations on fine-grained structural and semantic information of the objects in the scene. PRO consists of images containing stylized 2D objects that can be rendered starting from a relational representation, i.e., a graph, encoding their structure and style. Our contribution can be then summarized as follows:

- We provide a novel and general methodology to solve the task of generating images conditioned on an attributed graph that represents the structure and semantics of objects in the scene.

- We provide a specific implementation of such methodology, together with a learning procedure based on a task-independent pre-training to enable transfer to different problems.

- We propose a novel benchmark for image generation conditioned on relational representations of the objects in the scene.

Experimental results show that our approach compares favorably against non-relational baselines and that the proposed method is flexible and can be applied to different settings.

## 2 Preliminaries

A pose graph is a couple $\mathcal{G} = (\mathcal{V}, \mathcal{E})$, $\mathcal{V}$ is the set of vertices (or nodes) and $\mathcal{E}$ is the set of edges that connect such nodes. We define node attributes $\boldsymbol{v}_i = (\boldsymbol{p}_i, \boldsymbol{x}_i)$ where $\boldsymbol{p}_i \in \mathbb{R}^2$, $\boldsymbol{x}_i \in \mathbb{R}^d$ represents the 2D position of the $i$-th node in $\mathcal{V}$ and its $d$-dimensional feature vector, respectively. We denote with $\boldsymbol{V}$ the node attribute matrix $\boldsymbol{V} = (\boldsymbol{P} \| \boldsymbol{X})$ stacking all the node attribute vectors. The edge connecting the $i$-th to the $j$-th node is indicated as $e_{ij} = \langle i, j \rangle$. We assume the graph to be undirected and indicate its (symmetric) adjacency matrix as $\boldsymbol{A}$; nonetheless, our approach can be seamlessly extended to directed graphs. The described graph can be processed by stacking GNN layers; in particular, we rely on the *message passing* framework (Gilmer et al., 2017), which provides a template for building GNNs by specifying update and aggregation functions to process a node's representation and those of its neighbors.

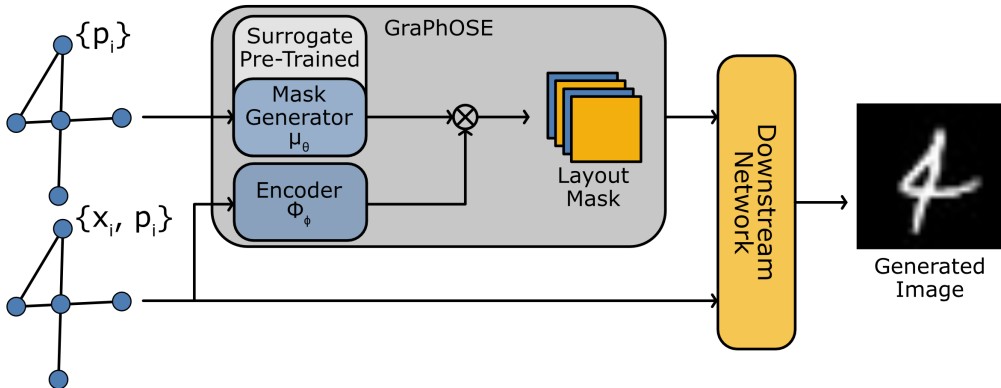

Figure 1: Our pipeline, with GraPhOSE in grey. $\mu_\theta$ gets pre-trained on surrogate masks. The downstream model, in yellow, can be any trainable generative model that accepts a 3-d tensor as conditioning input. The whole pipeline can be trained end-to-end.

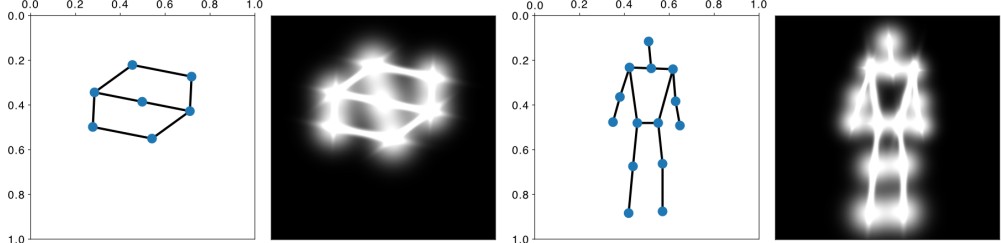

Figure 2: Surrogate mask for a random graph (left) and for a graph representing a person (right). Node positions in graph space are normalized between 0 and 1.

We indicate as *deep generative model* a neural network that, optionally given a conditioning input, can be trained to match a desired data distribution. The sampling is often obtained by feeding random noise from a fixed distribution to the model, while conditioning allows to control the generative process, e.g., by specifying a class label. In our case, we consider generative processes conditioned on the objects in the scene, represented as attributed pose graphs.

## 3 Graph-based Object-Centric Generation

In this Section, we first provide a high-level description of our method, consequently propose an architecture implementing the framework and finally present our surrogate pre-training objective.

### 3.1 Overview

Figure 1 provides a high-level view of our approach. Given an input pose graph $\mathcal{G}$, we wish to output a layout mask $\mathsf{L} \in [0,1]^{C \times H \times W}$, which will then be used to condition the downstream model on a target generative task. The mask generation process is designed to obtain the final mask by aggregating masks localized w.r.t. each node. In particular, the generation of $\mathsf{L}$ is decomposed into the generation of a mask $\boldsymbol{M}_i \in [0,1]^{H \times W}$ and a feature vector $\boldsymbol{f}_i \in [0,1]^{C}$ for each node $\boldsymbol{v}_i$, such that:

$$\mathsf{L} = \frac{1}{|\mathcal{V}|} \sum_{i \in \mathcal{V}} \boldsymbol{f}_i \otimes \boldsymbol{M}_i, \tag{1}$$

where $\otimes$ indicates the *tensor product*. Notably, each node mask, $\boldsymbol{M}_i$, is dependant on pose attributes, $\boldsymbol{P}$, only, while features $\boldsymbol{f}_i$ depend on both pose and attributes ($\boldsymbol{P} \| \boldsymbol{X}$), so that the former encodes only

structural information, while the latter will also eventually account for semantics. In particular, both $\boldsymbol{f}_i$ and $\boldsymbol{M}_i$ are computed by means of two learnable mappings, such that the core processing of GraPhOSE can be summarized as:

$$
\begin{aligned}
\boldsymbol{F} &= \Phi_\phi\left(\boldsymbol{V}, \boldsymbol{A}\right) \\
\mathsf{M} &= \mu_\theta\left(\boldsymbol{P}, \boldsymbol{A}\right) \\
\mathsf{L} &= \frac{1}{|\mathcal{V}|} \sum_{i \in \mathcal{V}} \boldsymbol{f}_i \otimes \boldsymbol{M}_i,
\end{aligned}
\tag{2}
$$

where $\boldsymbol{A}$ is the graph's adjacency matrix, $\mathsf{M}$ and $\boldsymbol{F}$ indicate respectively the tensor encompassing all the $\boldsymbol{M}_i$ masks and the matrix having as rows the node level representations $\boldsymbol{f}_i$. The $\Phi_\phi$ function can be learned end-to-end with the downstream model, as shown in Section 5. Differently, $\mu_\theta$ is pre-trained on a surrogate task designed to foster the learning of masks coherent with the structure of the object being generated. After pre-training, $\mu_\theta$ can be fine-tuned by being trained end-to-end with the downstream model on the target task. This approach is suitable to flexibly condition the generation of different objects based on their structure. Notably, we implement our method with two neural networks that operate in parallel: the *encoder* ($\Phi_\phi$) and the *mask generator* ($\mu_\theta$). The former implements function $\Phi_\phi$ from Equation (2), while the latter implements function $\mu_\theta$. The outputs of the two networks are combined as in Equation (2) to obtain the layout mask.

### 3.2 Implementing the encoder $\Phi_\phi$

The *encoder* ($\Phi_\phi$) network is based on what we call a *pose convolution* layer, which consists of the following message-passing operator:

$$
\boldsymbol{h}_i' = g_g\big(g_s\left(\boldsymbol{h}_i, \boldsymbol{p}_i\right) + \sum_{j \in \mathcal{N}(i)} g_l\left(\boldsymbol{h}_j, \boldsymbol{p}_j - \boldsymbol{p}_i\right), \boldsymbol{h}_i\big)
\tag{3}
$$

where $\mathcal{N}(i)$ is the set of neighbours of the $i$-th node, while $g_s$, $g_l$ and $g_g$ can be any learnable function (e.g., MLPs). Consistently with the previous naming convention, $\boldsymbol{h}_i$ and $\boldsymbol{p}_i$ indicate node features and position respectively. The layer is inspired by PointNet (Qi et al., 2017), but uses two distinct operators for processing the representation of the central node and computing messages coming from neighbors. In particular, $g_s$ can be seen as implementing a parametrized skip connection to mitigate over-smoothing node representations (Li et al., 2019). The final node encodings are obtained by stacking blocks of the form

$$
\mathcal{G}_{\boldsymbol{H}'\|\boldsymbol{P}} = \mathrm{BN}\left(\mathrm{RELU}\left(\mathrm{PCONV}\left(\mathcal{G}_{\boldsymbol{H}\|\boldsymbol{P}}\right)\right)\right),
\tag{4}
$$

which is a common way of chaining processing layers in deep neural networks. In particular, BN denotes *batch normalization* (Ioffe & Szegedy, 2015), RELU is the activation function, PCONV is our pose convolution layer and $\mathcal{G}_{\boldsymbol{H}\|\boldsymbol{P}}$ a pose graph with features $\boldsymbol{H}$ and node coordinates $\boldsymbol{P}$.

### 3.3 Implementing the mask generator $\mu_\theta$

The *mask generator* ($\mu_\theta$) network consists of a first block of stacked pose convolutional layers analogous to the ones in Equation (4). The outputs of these layers are then reshaped into bi-dimensional matrices used as input for the second stack of layers consisting of a combination of shared 2D convolution blocks, interleaved with relational *pose convolution 2D* layers. Such layers implement the following message-passing function:

$$
\begin{aligned}
\boldsymbol{O}_i &= \sum_{j \in \mathcal{N}(i)} g_o\left(\boldsymbol{H}_i, \boldsymbol{H}_j\right) \\
\boldsymbol{H}_i' &= g_s\left(\boldsymbol{H}_i\right) + \sigma\left(\boldsymbol{O}_i\right) \odot g_g\left(\boldsymbol{O}_i\right)
\end{aligned}
\tag{5}
$$

where, $\odot$ denotes the Hadamard product, $\mathcal{N}(i)$ is the set of neighbors of node $i$ and $g_s$, $g_l$ and $g_g$ can be any learnable function that has two-dimensional inputs and outputs.

In our case, $g_g$ is a 2D convolutional layer while $g_o$ is a convolutional block with upsampling and skip-connection and $g_s$ is a linear upsampling operation followed by a 2D convolution, where the upsampling is needed to match the dimensions of the output of $g_o$. The rationale behind the design of Equation (5) is promoting heterogeneity between the masks generated by different nodes. Notably, $g_s$ can act as a skip connection preserving the node representations while the gating operation allows for selectively aggregating information coming from neighbors. Indeed, over-smoothing the node features would be particularly detrimental as it would compromise the locality of the masks learned w.r.t. each node. Said property is desirable as, per Equation (1) this would in turn make the learned node features, $\boldsymbol{f}_i$, localized w.r.t. their spatial neighborhood. Note that these soft constraints, i.e., architectural biases, can be seen as a form of regularization aligned with object-centric generation tasks.

### 3.4 Surrogate Task

As mentioned previously, we want to pre-train GraPhOSE, in particular $\mu_\theta$, on the generic surrogate task of mapping a pose graph to a 2D mask. For this purpose, we define the surrogate mask associated with a pose graph $\mathcal{G}$, as shown in Figure 2, which, intuitively, is a grayscale image that depicts the structure of the graph.

The mask for the whole graph is obtained by composing partial masks relative to each node and edge. In particular, given pixel coordinates $c$, we define the value of each $i$-th partial surrogate mask at that pixel as

$$\mathrm{S}_{\boldsymbol{N},i}\left(\boldsymbol{c}\right) = \exp\left(-\frac{\|\boldsymbol{p}_i - \boldsymbol{c}\|_2^2}{2\sigma^2}\right), \tag{6}$$

and, analogously, the value of the partial surrogate mask associated with each $e_{ij}$ edge as

$$\mathrm{S}_{\boldsymbol{E},ij}\left(\boldsymbol{c}\right) = \sqrt{\frac{\exp\left(-d_{ij}(\boldsymbol{c})^T \cdot \boldsymbol{T}_{ij}^{-1} \cdot d_{ij}(\boldsymbol{c})\right)}{\left(2\pi\right)^2 \cdot \det\left(\boldsymbol{T}_{ij}\right)}}, \tag{7}$$

where $\boldsymbol{T}_{ij}$ denotes a $2 \times 2$ rotation and scaling matrix dependent on the length and orientation of the segment connecting the $i$-th and $j$-th nodes (see the supplementary material for the details). Conversely, $d_{ij}(\boldsymbol{c})$ is defined as

$$d_{ij}(\boldsymbol{c}) = \boldsymbol{c} - \left(\frac{\boldsymbol{p}_i + \boldsymbol{p}_j}{2}\right) \tag{8}$$

and denotes the distance between $\boldsymbol{c}$ and the midpoint between the coordinates of nodes $i$ and $j$.

The final surrogate mask is obtained, for each pixel $c$, as

$$\mathrm{S}_{\mathcal{G}}\left(\boldsymbol{c}\right) = \sum_{i \in \mathcal{G}} \mathrm{S}_{\boldsymbol{N},i}\left(\boldsymbol{c}\right) + \sum_{\langle i,j \rangle \in \mathcal{G}} \mathrm{S}_{\boldsymbol{E},i,j}\left(\boldsymbol{c}\right) \tag{9}$$

which is the pixel-wise sum of the masks associated with each node and edge. All the values are then clipped between 0 and 1. More details about the computation of the mask can be found in Appendix A.3. Intuitively, the mask corresponding to a node is obtained by considering an isotropic bi-variate gaussian centered into the node's coordinates and with standard deviation $\sigma$. The mask corresponding to an edge, instead, is obtained by considering a bi-variate gaussian centered w.r.t. the midpoint between the edge's vertices, and a covariance matrix dependent on the distance between the two points and the orientation of the line joining them. The surrogate mask obtained in such a way is agnostic w.r.t. the object represented by the graph and hence general. Note that differently from, e.g., segmentation masks, the mask we are referring to depends entirely on the structure of the objects in the image being generated and not on their class.

As a final remark, the surrogate mask has a lattice structure, which may not properly resemble the desired mask for all target tasks; indeed, depending on the specific object, some loops should be filled and the silhouette of each specific part refined. Nonetheless, such surrogate masks are helpful in providing supervision for the pre-training routine and act as a regularization for the model. Details on the fine-tuning procedure on downstream tasks are discussed in Section 6.1. The pre-training routine is carried out by minimizing a surrogate loss $\mathcal{L}_{re}$ based on a reconstruction error, e.g., mean squared error or binary cross entropy.

## 4 Related Works

Image generation is a popular application of deep learning models from *generative adversarial networks* (GANs) (Goodfellow et al., 2014; Gui et al., 2021), to *variational autoencoders* (VAEs) (Kingma & Welling, 2014; Vahdat & Kautz, 2020; Park et al., 2020) and, more recently, *diffusion models* (Sohl-Dickstein et al., 2015; Song & Ermon, 2019; Ho et al., 2020). Concurrently, many researchers have explored ways of conditioning the generation of such images, from simple class labels (Mirza & Osindero, 2014; Brock et al., 2019), to fully articulated sentences (Ramesh et al., 2022; Rombach et al., 2022) or even other images (Isola et al., 2017; Liu et al., 2017; Zhu et al., 2017). Although no previous work directly exploits pose graphs to guide image generation, several approaches focused on *scene-graph-conditioned image generation* and *pose-conditioned image generation*. Even though these tasks present some similarities with our work, they are fundamentally different, as discussed in the remainder of this section.

**Scene-Graph-Conditioned Image Generation**  Scene graphs are a way of representing a visual scene context as a graph linking object nodes through predicate edges. These graphs are obtained by parsing natural language sentences (e.g., the phrase *A man sits on a bench* turns into a graph with two nodes with attributes *man* and *bench*, respectively, linked by an edge with attribute *to sit*), making these approaches similar to text-based image generation. The first method for conditioning image generation on scene graphs was introduced by Johnson et al. (2018), which propose a GNN mapping the scene graph into a scene layout used to bias a generator network. Ivgi et al. (2021) improved such an approach by designing a GNN that directly operates on 2D feature maps. Differently from the above methods, however, our formulation represents objects as graphs of related parts, with both structural and semantic information; scene graphs contain only semantics and represent one object per node, as they are derived from natural language parsing. This makes such methods unsuitable for our problem setting.

**Pose-Conditioned Image Generation**  Several prior works condition image generation on the pose of the objects being generated. In particular, several methods deal with images from human poses, with Ma et al. (2017) introducing the first deep learning approach based on conditioning a reference image on a target pose. The method introduced by Ma et al. (2017) has been thereafter extended by implementing deformable convolutions (Siarohin et al., 2018), adopting dense poses (Neverova et al., 2018) and global style attributes (Men et al., 2020) as conditioning. Moreover, it has been used to address the problem of person re-identification (Qian et al., 2018). Horiuchi et al. (2021) exploits a graph-based representation for the pose paired with a reference image encoding semantics. To the best of our knowledge, none of the previous works jointly encode pose and semantics information in a graph and then use such graph as an architectural bias for the neural processing. Exploiting this previously unused relational information allows our method to compare favorably against the state of the art w.r.t. flexibility, generality, and effectiveness. Moreover, note that this family of techniques solves an image-to-image task, conditioned on the pose (i.e., given a person's image as input, the model outputs an image with the same context, where the person matches the target pose). This is a very different problem w.r.t. to the graph-to-image task we are tackling. While a direct comparison would not be meaningful, classical pose-conditioned approaches can be better suited to tasks where the conditioning is a template image that needs to be modified or augmented.

## 5 Experiments

We start the empirical validation of the proposed method by focusing on the pre-training of the mask generator $\mu_\theta$ with surrogate masks created from procedurally generated graphs. Then, we introduce relevant baselines for our model and compare them on image generation tasks w.r.t. both synthetic and real-world data. We carried out our experiments with a GAN, even though the same principles can be extended to work with other methodologies. For this, noise is sampled at the input graph level, such that, $\boldsymbol{V} = (\boldsymbol{P} \| \boldsymbol{X} \| \boldsymbol{Z})$ with $\boldsymbol{Z} \in \mathbb{R}^{N \times Z} \sim \mathcal{N}(0, 1)$. The code used to run the experiments is provided in the supplementary materials.

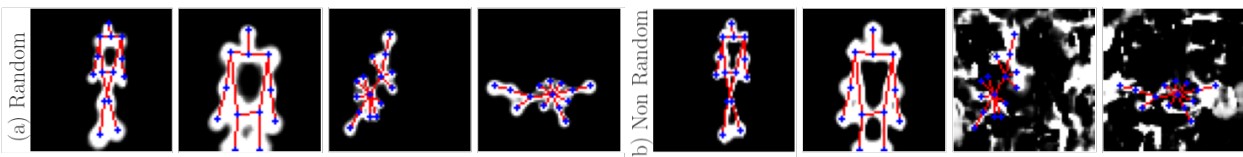

Figure 3: Sample of generated masks. For each one, the large figure is the aggregated mask, while the small ones are those associated with each node. The blue dots highlight the position of the accounted node. (a) is a random graph like those used for pre-training; (b) and (c) are simple handcrafted ones.

Figure 4: Masks generated by pre-training on random graphs (a) and on the Humans task's ones (b). For each group, the first two columns are samples from the Humans task, last two are random. In (b), performance clearly degrades out of distribution.

## 5.1  Surrogate Pre-Training

For surrogate pre-training, we generate random graphs using the *Barbasi-Albert* (Albert & Barabási, 2002) or a *Erdos-Renyi* (ERDdS & R&wi, 1959; Gilbert, 1959) model, with a number of nodes in the interval [5, 30]. The position of each node is determined by using *Kamada-Kawai*'s force-directed graph drawing algorithm (Kamada & Kawai, 1989). These choices aim at generating (almost planar) graphs that can easily be drawn on the plane, i.e., with only a few intersections among edges. More detailed information on the setup of the generators are provided in the supplementary materials. Target masks are computed as in Equation (9) and the model is fitted to minimize the *binary cross entropy* (BCE) loss. Figure 3 shows sample results of the mask generation process. Each one of the larger images corresponds to the full mask generated from the input graph (shown in blue and red), while the smaller images show masks relative to each node (in blue). The model produces masks that match the topology of the graph, even for complex structures. More interestingly, results show that the proposed architecture is indeed capable of generating localized node-level masks that capture the structure of the neighborhood of each node. By looking at samples (b) and (d), which contain handcrafted graphs, we see that this property is preserved for graphs with a small diameter; even though, nodes with high connectivity tend to produce richer masks that may span it completely. Indeed, locality is preserved against over-smoothing which is typical of isotropic graph convolutional filters; conversely, the learned node-level masks are diverse and properly localized. Figure 4 additionally shows how pre-training on randomly generated graphs can act as regularization by yielding a model able to perform properly for graphs outside the distribution of the downstream task.

## 5.2  Baselines

Comparing existing models to ours is not straightforward. As detailed in Section 4, these methods lack flexibility concerning the objects they can generate and the representation of structural and semantic constraints. Moreover, the tasks they address fundamentally differ from ours, as they rely on different inputs, and thus they cannot be directly compared to our approach. As described in Section 3, GraPhOSE learns to map the graph into a layout mask by learning both node representations and node masks. We thus compare it to two baselines: both use a fixed (non-learnable) mapping to generate the node masks; the first (GNN Conditioner) uses a GNN-based encoder to learn node representations, while the second (FNN Conditioner) uses a feedforward neural network (FNN) based one (i.e., it does not use relational information). Considering Equation (2), we can contextually schematize the baselines as computing mask **M** as

$$\mathbf{M} = \|_{i \in \mathcal{V}}\big(\mathrm{S}_{\boldsymbol{N},i}\left(\boldsymbol{C}\right) + \sum_{j \in \mathcal{N}(i)} \mathrm{S}_{\boldsymbol{E},ij}\left(\boldsymbol{C}\right)\big) \tag{10}$$

and, for the FNN-based one only, as computing $\boldsymbol{F}$ as

$$\boldsymbol{F} = \Psi_\psi(\boldsymbol{V}) \tag{11}$$

where $\|_{i \in \mathcal{V}}(\cdot)$ indicates concatenation along a new axis, while, for ease of notation, $\mathrm{S}_{\boldsymbol{N},i}(\boldsymbol{C})$ and $\mathrm{S}_{\boldsymbol{E},ij}(\boldsymbol{C})$ denote operations applied over all the pixel coordinates w.r.t. nodes and edges, respectively. Note that the FNN Conditioner has still access to structural information through the fixed masks which makes the comparison against GraPhOSE meaningful. For all the baselines (including GraPhOSE), we use the same downstream model based on a simple stack of layers with 2D Convolutions, batch normalization (Ioffe & Szegedy, 2015) and ReLU activation and residual connections (He et al., 2016). Moreover, similarly to Johnson et al. (2018), each layer may take as input both the previous layers's output and the downsampled layout mask to condition the generative process at the different processing steps. The discriminator is based on similar building blocks and receives the graph as input, together with the real/fake image. See Appendix A.7 for more details on the implementation for each baseline. For GraPhOSE we pre-train the mask generator as described in Section 5.1, and then train the whole model on the downstream task. Note that, during the end-to-end training, we drop the reconstruction loss originally used during the pre-training. This is to allow for adapting the masking mechanism to the downstream task, removing any bias coming from the surrogate loss. All the baselines were trained under the same settings; more details can be found in Appendix A.2.

### 5.3 Datasets

As previously mentioned we consider 2 benchmarks based on synthetic and real-world data. We explicitly design the synthetic dataset, named PRO, to highlight the benefits of object-centric relational representations. PRO consists of images containing simple stylized objects that can be rendered starting from a pose representation in which each keypoint has class and style attributes, i.e., from a relational description of the image. Different objects are represented at different degrees of abstraction w.r.t. their appearance to make the task of generating objects, with a coherent style, from the representation more challenging. The style and visual appearance of the different rendered components vary smoothly with respect to the structure of each object (as it is often the case in the real world), making the adoption of appropriate inductive biases particularly appealing. For our experiments, we generated 100000 samples. For what concerns the second task, referred to as *Humans*, it consists of the problem of generating images of humans from key-points with class attributes (e.g., ankle, shoulder). To build a dataset of examples, we leveraged the existing MPII Human Pose (Andriluka et al., 2014), Market 1501 (Zheng et al., 2015) and DeepFashion (Liu et al., 2016) datasets. For the DeepFashion dataset, we use just the Fashion Landmark Detection and the In-Shop Retrieval subsets. The keypoint annotations for DeepFashion In-Shop Retrieval and Market 1501 are based on Zhu et al. (2019), and were generated by using the open-source software OpenPose (Cao et al., 2019). For DeepFashion Fashion Landmark we instead rely on MediaPipe's BlazePose (Bazarevsky et al., 2020), while MPII Human Pose already contains pose features. All annotations are reduced to the COCO keypoint standard (Lin et al., 2014) and used to generate the corresponding graphs with node coordinates normalized between 0 and 1, resulting in a dataset of roughly 300000 samples. Images from the real-world datasets have been downscaled (or padded) to a $64 \times 64$ shape. Images in PRO are generated with size $128 \times 128$ to ensure that objects are not too small. Further details on the datasets are provided in Appendix A.5 and A.6.

## 6 Results

In the following section, we discuss the results obtained on the synthetic and real-world datasets. On the former, we used a small downstream model with around $28K$ parameters, while, on the latter, a more complex one, with roughly $5M$ parameters, was employed; details are provided in the appendix. Moreover, for the synthetic dataset, models that leverage relational representations receive inputs with 10% of the semantic node attributes randomly masked in 50% of the samples; we do so to show that structures can be used to generate coherent images from partial conditioning. Table 1 shows the Frechet Inception Distance (FID) (Heusel et al., 2017) and Inception Score (IS) (Salimans et al., 2016) for our model and the baselines w.r.t. both datasets, together with the Structural Similarity Index Measure (SSIM) (Wang et al., 2004) for

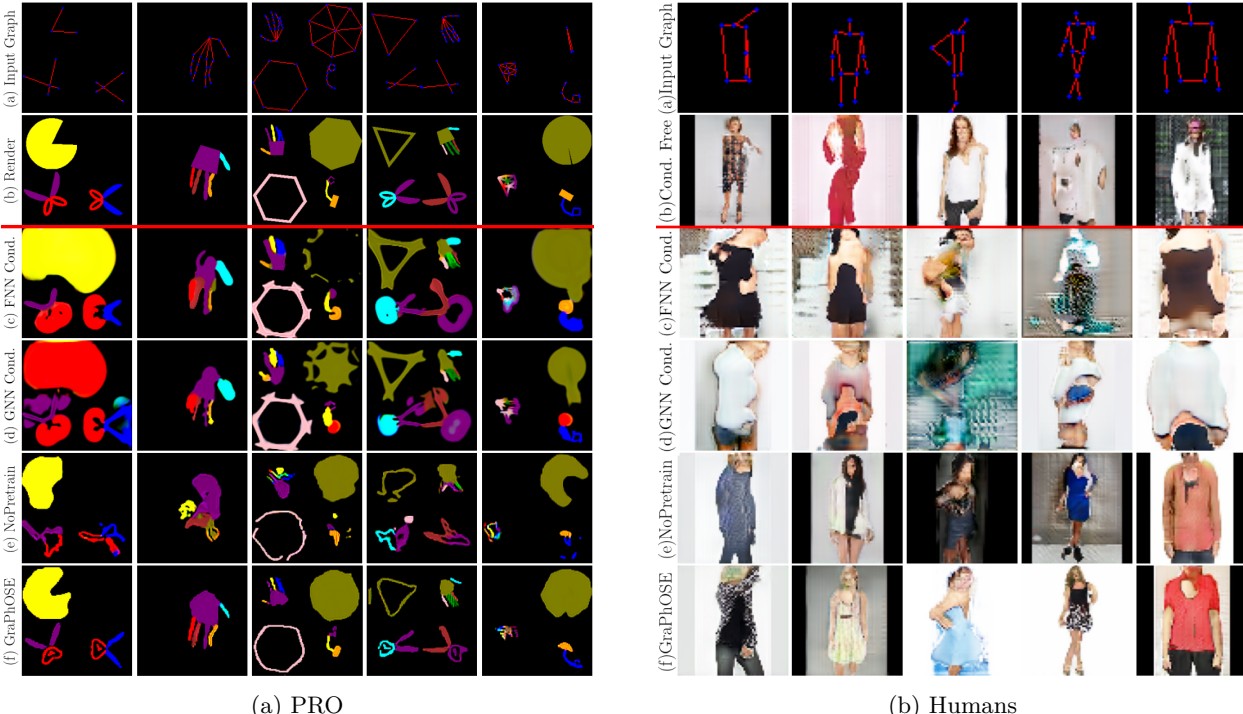

(a) PRO                                          (b) Humans

Figure 5: Sample results for PRO (left) and Humans (right) tasks. Row (a) is the input graph. Generated samples come from: the PRO exact renderer (b - left), the unconditioned downstream model (b - right), the FNN-based baseline (c), the GNN-based one (d), GraPhOSE without pre-training (e), GraPhOSE (f).

Table 1: Frechet Inception Distance (FID) (lower is better) and Inception Score (IS) (higher is better) for PRO and Humans datasets. Also the Structural Similarity Index Measure (SSIM) is reported for the PRO dataset.

| MODELS | PRO | | | Humans | |
|---|---|---|---|---|---|
| | FID ↓ | IS ↑ | SSIM ↑ | FID ↓ | IS ↑ |
| FNN Conditioner | $103.86_{\pm15.93}$ | $3.23_{\pm0.03}$ | $0.74_{\pm0.05}$ | $58.91_{\pm3.77}$ | $3.52_{\pm0.10}$ |
| GNN Conditioner | $105.23_{\pm9.72}$ | $3.24_{\pm0.02}$ | $0.78_{\pm0.02}$ | $118.18_{\pm78.54}$ | $3.58_{\pm0.16}$ |
| GraPhOSE No-Pretrain | $90.9_{\pm6.71}$ | $3.11_{\pm0.02}$ | $0.77_{\pm0.01}$ | $125.22_{\pm84.14}$ | $3.52_{\pm0.27}$ |
| **GraPhOSE** | $\mathbf{66.5_{\pm4.82}}$ | $2.92_{\pm0.01}$ | $\mathbf{0.84_{\pm0.01}}$ | $\mathbf{54.46_{\pm7.57}}$ | $3.41_{\pm0.12}$ |

the PRO dataset. GraPhOSE achieves a significantly better FID and SSIM score compared to the baselines. Indeed, in the PRO dataset, GraPhOSE outperforms the baseline both with and without pre-training for the FID measure. Note that SSIM measures the accuracy of image reconstruction, making it a meaningful metric only for the PRO dataset, where the input graph fully describes the target image. For the human pose dataset, pre-training acts as regularization allowing for reducing the variance among runs. As expected, the IS, which evaluates the distinctiveness and diversity of the generated samples, does not highlight significant differences among the models, as these properties are not highly influenced by their architectural differences. We can then assess compliance w.r.t. the conditioning visually, in a qualitative manner.

In this regard, Figures 5a and 5b, show results generated, by the compared models, for the two tasks. We can see that, in PRO, GraPhOSE, row (f), is clearly superior to the other baselines, in particular when the surrogate mask further deviates from the object's visual appearance. The generation quality is consistent across different numbers of objects and scales, showing the approach is suitable to handle a wide range of

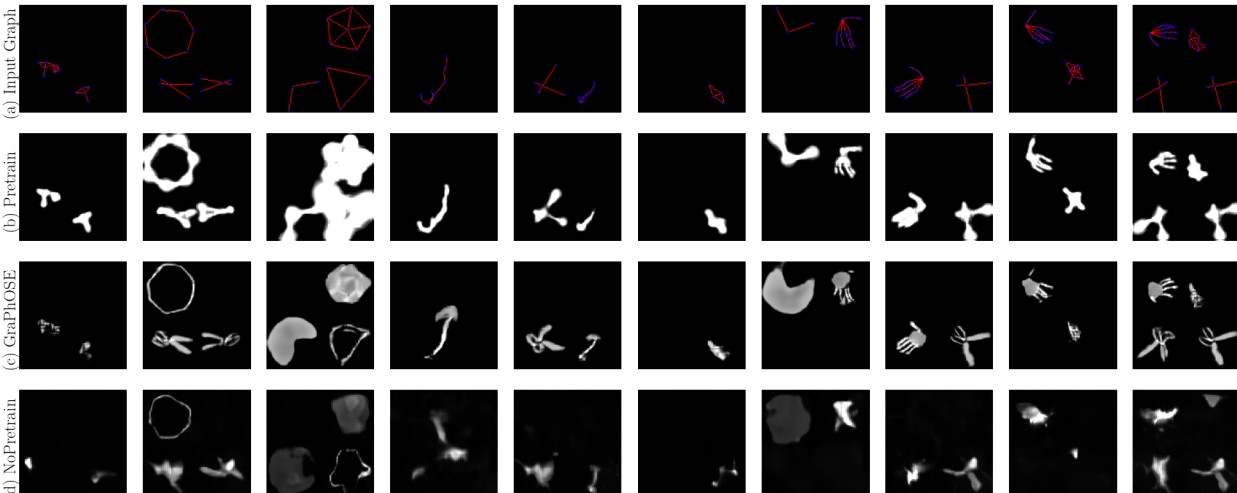

Figure 6: Masks generated by GraPhOSE after pre-training (b) and after subsequent training on the downstream task (c). (d), shows those generated by skipping pre-training, while (a) is the input graph

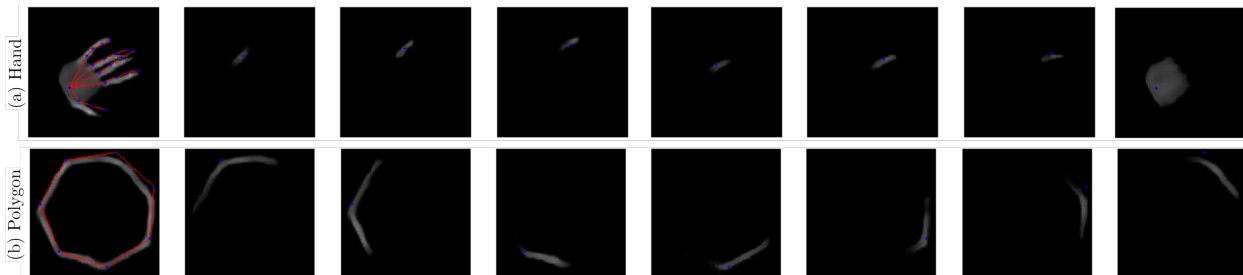

Figure 7: Samples of individual node masks generated from GraPhOSE. The first column is the cumulative mask for the whole input graph (superimposed with nodes in blue and edges in red). Individual node masks have the reference node highlighted in blue. Hand masks (a) are only a sub-sample for ease of visualization.

possible scenes. Moreover, the FNN Conditioner is not capable of handling missing attributes or nodes, which hinders its flexibility, in particular for real-world settings. For Humans, we again see that images generated by GraPhOSE have better visual appeal and compliance with the input pose, even when not all the keypoints are present in the conditioning graph. Furthermore, notice how images from the FNN Conditioner, row (c), are visually worse than those from models with a higher (worse) FID score. In general, relational representation plays a significant role in guiding the generation toward a coherent result, as structure alone, i.e., even without semantics, can be leveraged to characterize silhouettes, and neighboring information can help reconstruct missing features. Examples of this behavior are shown in Appendix B.1. Note that the generative models used in this work are purposely simple, as our experiments focus on the conditioning capabilities rather than image quality (i.e., detail, realism). The resulting image quality is comparable to that of architectures that use similar generators (Johnson et al., 2018; Yan et al., 2016). Images of better quality can be obtained at the expense of increased computational and sample complexity by adopting more complex generators trained at higher resolutions.

## 6.1 Task specific mask adaptation

The purpose of pre-training on surrogate masks of random pose graphs is to learn a general mapping between the graph and the image representation of the structure it entails. However, this learned mapping might not produce the exact visual properties desirable for a specific downstream task. We wish to adapt to these properties while learning the target task, during the end-to-end training. Figure 6 shows how this indeed

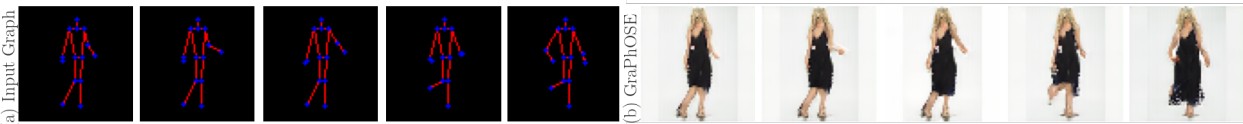

Figure 8: Results (b) of changing input nodes' position (a) while maintaining other inputs constant.

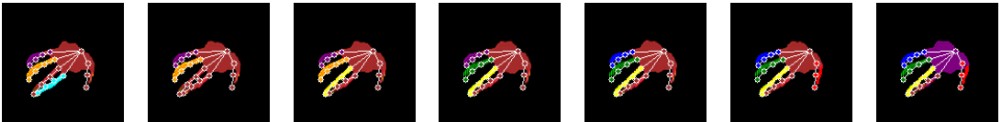

Figure 9: Results of changing the input nodes' semantic attributes (i.e. color), for samples of the PRO dataset. The input graph, with according colored nodes, is superimposed onto the generated image. Nodes belonging to the same finger change color in group as otherwise the input graph would be out of distribution w.r.t. training data.

happens for the masks being generated before (b) and after (c) training on a target task. This adaptability is particularly relevant, as it shows that after pre-training the mask generator towards the correct direction, we can have it learn the specifics of the target downstream task. Differently, the images generated by training GraPhOSE without pre-training on surrogate masks (d) do not visually resemble the input pose. These results suggest the lack of structure in the mask leads to degradation of the performance, in particular when the target data distribution is not heavily biased towards a few poses (e.g., Humans contains mostly poses of people standing). Note that, even without pre-training, the model still leverages the relational inductive biases encoded by the input graphs and accounted for by our model architecture. This shows the usefulness of pre-training as a soft regularization, which localizes node features w.r.t. corresponding portions of the image. Further examples of this property are shown in the appendix. Figure 7 shows how individual node masks preserve a property of locality after training on the downstream task, even though no particular constraint, aside from architectural biases, enforces this. These results are really meaningful, as localized masks allow, by GraPhOSE's design, for better local conditioning on node level semantics. Moreover, it entails the learning of a distributed representation, where each node describes a specific part, rather than a collapsed representation in which all nodes contain a representation of the whole object. Further examples of this node-mask locality can be found in Appendix B.2.

## 6.2 Structure and style sensitivity analysis

To assess whether the model is able to disentangle structure and style, we experiment with providing it a series of human graphs with slightly different poses while fixing all the remaining attributes of the conditioning. As shown in Figure 8, generated figures are rearranged according to the changes in the input pose while the style is mostly unchanged. Note that, however, in case of more significant changes in the pose, the resulting images can be visually different. This emergent property is nonetheless interesting and provides ground for future research. Similarly, we assess the sensitivity of the method to incremental perturbations of the semantic node attributes for samples of the PRO dataset. Example results are shown in Figure 9. We can see that the generated object's structure is not affected by the semantic changes. Only the colors change according to the changes in node attributes. This further highlights the disentanglement between structural and style control over the generated images.

## 7 Conclusions and Future Work

We propose GraPhOSE, a method to exploit attributed graphs as object-centric relational inductive biases to flexibly condition image generation. Our approach has several advantages compared to standard approaches which result in a scalable and flexible framework. Notably, we shone a light on the properties of relational representation in this context, by showing how they can be used to regularize and manipulate the generative

model. We also provided a novel synthetic dataset that pairs images with their object-centric relational representation, introducing a benchmark to support further studies. Future research might target approaches to generate meaningful and coherent masks without relying on pre-training, or investigate methods to disentangle the different elements that contribute to the final generated image. We argue that our study has the potential of sparking new interest in object-centric relational representations for generative modeling and that the results presented here constitute the first building block for future research in this direction.

**Acknowledgments**

The authors wish to thank *Daniele Zambon* for the helpful suggestions provided during the development of this work.

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

# A    Implementation Details

In the following we clarify some details on the settings used to train the models, on the specifics of some models' architectures, as well as further details on surrogate masks and the hardware and software we used to run the experiments.

## A.1    Hardware and Software

The code to reproduce the experiments and generate the PRO dataset is available online[1]. It is all written in *Python 3.9*, leveraging *Pytorch* (Paszke et al., 2019) and *Pytorch Lightning* (Falcon & team, 2019) to define models and data, while *Hydra* (Yadan, 2019) is used to manage the experiment configuration. *Weights & Biases* (Biewald, 2020) is used to log and compare experiment results. We run all the experiments on an *NVIDIA RTX A5000* GPU equipped with 24GBs of VRAM.

## A.2    Training settings

Each model is trained under the same settings and metrics are computed over three different runs each. In particular, we use the *Adam* (Kingma & Ba, 2015) optimizer, with *cosine annealing* (Loshchilov & Hutter, 2017) learning rate schedule with period 300 for the PRO dataset and 150 for the Humans one, starting learning rate 0.002 and final learning rate 0.00002. Models are trained up to 300 (PRO) or 150 (Humans) epochs each, with a batch size of 64 and early stopping with patience 50 on the FID. Notice that the learning rate for the parameters of the pre-trained mask generator used in GraPhOSE was reduced to $\frac{1}{2}$ (PRO) or $\frac{1}{100}$ (Humans) with respect to the learning rate of the other parameters, in order to avoid catastrophic foregetting at the beginning of training. This procedure could be enhanced by slowly raising the learning rate of the pre-trained parameters back to the value of other parameters, as training epochs go by, with the aim of favoring mask adaptation after the training has stabilized during the first epochs. The reduction was much stronger for Humans in order to counter balance the heavy bias the dataset had towards poses of people standing up with arms straight down. However, this reduction factor is indeed an hyper-parameter that can be effectively used to allow for more or less deviation from the surrogate masks, depending on the downstream task characteristics.

## A.3    Surrogate Mask

Referring to Equation (7), we explicitly define $\boldsymbol{T}_{ij}$ as

$$
\begin{aligned}
d_{ij} &= \frac{\|\boldsymbol{p}_i - \boldsymbol{p}_j\|_2^2}{4} \\
\alpha_{ij} &= \arctan 2\left(\boldsymbol{p}_j - \boldsymbol{p}_i\right) \\
t_{a,ij} &= d_{ij} \cdot \cos\left(\alpha_{ij}\right)^2 + \frac{d_{ij}}{a^2} \cdot \sin\left(\alpha_{ij}\right)^2 \\
t_{b,ij} &= d_{ij} \cdot \sin\left(\alpha_{ij}\right)^2 + \frac{d_{ij}}{a^2} \cdot \cos\left(\alpha_{ij}\right)^2 \\
t_{c,ij} &= \left(d_{ij} - \frac{d_{ij}}{a^2}\right) \cdot \sin\left(\alpha_{ij}\right) \cdot \cos\left(\alpha_{ij}\right) \\
\boldsymbol{T}_{ij} &= \begin{bmatrix} t_{a,ij} & t_{c,ij} \\ t_{c,ij} & t_{b,ij} \end{bmatrix},
\end{aligned}
\tag{12}
$$

where vectors $\boldsymbol{p}$ denote node coordinates, $\arctan 2$ is the element-wise 2-argument arctangent and $a$ denotes a parameter that regulates the scaling ratio of the second dimension w.r.t. the first one. In our experiments $a$ is set to 10.

---

[1]https://github.com/LucaButera/graphose_ocrrig

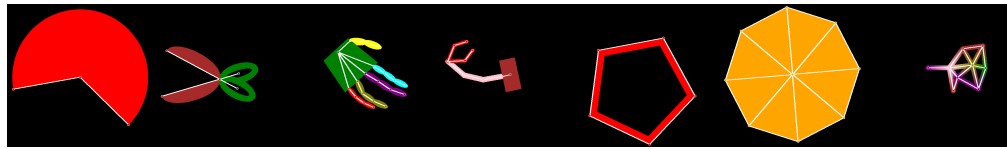

Figure 10: Examples of objects present in the PRO dataset. From left to right: Pie, Scissors, Hand, Robotic Arm, Hollow Polygon, Filled Polygon, Lattice

### A.4 Random Graphs for Pre-Training

Referring to Section 5.1, we set the *Barbasi-Albert* model's number of edges for new nodes parameter to 1 in 90% of the cases and to 2 in the remaining 10%, if the number of nodes is less than 10. For graphs with more than 10 nodes, it is always set to 1. *Erdos-Renyi* model's edge probability is set to $\exp\left(\frac{1}{n}\right) - 0.95$, where $n$ is the number of nodes.

### A.5 PRO Dataset

Our synthetic *Pose-Representable Objects (PRO)* dataset consists of square images (i.e., with same height and width) rendered from random graphs that represent certain object's parts and their color. A graph representing an object has a fixed connectivity (i.e., adjacency matrix) and each node has color (from a pool of possible ones) and class attributes together with a coordinate in 2D space. Coordinates and colors, for each node in a sample, are generated randomly within reasonable boundaries (e.g. to stay within the image or to comply to a specific structure, like that of a regular polygon). Figure 10 shows some examples of each object type in our dataset; the graph that describes the object has the edges superimposed in white, while the nodes are colored accordingly to their attributes. Images from the *PRO* dataset can contain multiple objects, however we experimented with up to four objects for image, in order to keep their size reasonably big. Regarding the objects' structural constraints, we highlight the following properties: *Pie* has an inner angle of up to 135°, as the two pie tips have the same class and it would not be possible to discriminate which part of the pie has to be filled otherwise. Pie nodes have always the same color. *Scissors* have the structural constraint that the angle between the blades and the handles must be the same and be contained between 30° and 90°. Also blades and handles must respectively have the same color. For the *Hand* object we have constraints on the maximum relative angle between each phalanx, in order to avoid impossible orientations; moreover, fingers must have a consistent color. Hands can uniformly be either "right" or "left" hands. The *Robotic Arm*, similar to the Hand, has constraints on the relative angles between its parts, to similarly avoid collapsed configurations. Additionally, the number of segments in the arm can vary between 3 and 7 and their color must not change. Similarly, also the prongs must have a consistent coloring. *Hollow Polygon* and *Filled Polygon* must always have all nodes of the same color and the presence of a "center" node determines if the polygon is filled or not. We experiment with polygons up to 8 vertices. The *Lattice* is the only structure with randomized connectivity, as nodes, uniformly distributed between 3 and 9, are sampled from a Poisson Disk distribution (Cook, 1986) and are connected if their distance is less than 20% of the whole image width. The lattice is also the only object in which each node has a random color and the segment connecting two nodes is colored with a linear gradient going from one node's color to the other's.

Table 2 contains a summary of the classes for each object's node. We would like to underline that the presence of structure in the assignment of colors and classes to object nodes is not just to make them visually consistent, but is fundamental in order to make it possible to reconstruct missing attributes by using relational information and acquired knowledge on how said structure behaves.

Regarding dataset size, in our experiments we generated 100000 images. We employed 1000 of these as validation set, and another 1000 as test set. Images were sampled so that the dataset contains a uniform distribution of different objects. Moreover, also the number of objects that appear in an image at once, between 1 and 4, is uniformly distributed across the dataset. Train, validation and test splits, while random, preserved these uniform distributions.

Table 2: Node classes for each object.

| Object | Classes |
|--------|---------|
| Pie | center, tip |
| Scissors | pivot, blade, handle |
| Hand | wrist, finger |
| Robotic Arm | base, arm, prong |
| Hollow Polygon | vertex |
| Filled Polygon | vertex, p_center |
| Lattice | l_vertex |

Table 3: The Encoder $\Phi_\phi$.

| Layer | H-params | Input | Output |
|-------|----------|-------|--------|
| MultiEmbedding | $8, cat$ | $\boldsymbol{X}$ | $\boldsymbol{O}$ |
| PoseConv | $8, 8, 8$ | $\boldsymbol{O}\|\boldsymbol{Z}, \boldsymbol{P}, \boldsymbol{A}$ | $\boldsymbol{O}$ |
| BatchNorm | | $\boldsymbol{O}$ | $\boldsymbol{O}$ |
| ReLU | | $\boldsymbol{O}$ | $\boldsymbol{O}$ |
| PoseConv | $8, 16, 16$ | $\boldsymbol{O}, \boldsymbol{P}, \boldsymbol{A}$ | $\boldsymbol{O}$ |
| BatchNorm | | $\boldsymbol{O}$ | $\boldsymbol{O}$ |
| ReLU | | $\boldsymbol{O}$ | $\boldsymbol{O}$ |
| PoseConv | $16, 32, 32$ | $\boldsymbol{O}, \boldsymbol{P}, \boldsymbol{A}$ | $\boldsymbol{O}$ |
| Sigmoid | | $\boldsymbol{O}$ | $\boldsymbol{O}$ |

## A.6  Humans Dataset

The Humans dataset consists of roughly 300000 images, coming from MPII Human Pose (Andriluka et al., 2014), Deep Fashion (Liu et al., 2016) and Market 1501 (Zheng et al., 2015), which are all widely adopted benchmarks for pose estimation and other human body related tasks. Out of these, around 3000 images, were used for validation and an analogous size for testing. Train, validation, and test sets were partitioned at random but the proportion of samples from the 3 original benchmarks was maintained. Within the Humans task, the only object type is *Person*, while there are 14 node classes (left and right shoulders, elbows, wrists, ankles, knees, hips plus base of the neck and top of the head). Note that, differently from the PRO dataset, these real-world images contain different backgrounds and lighting conditions. We do not address the explicit conditioning over these properties in our model, however these could be incorporated via a vector representation that acts as a global graph attribute.

## A.7  Architectures

In the following we give a more precise specification for the architecture we used in our experiments, while specifying that these represent only a possible way of implementing our approach and deeper or more complex architectures might be better suited for different tasks. Coherently with previous notation, Tables 3 to 7 describe the architectures of the different components in our framework; while Tables 8 and 9, for clarity, describe recurring building blocks (i.e., used multiple times) inspired by Brock et al. (2019). In particular, we clarify the meaning of some layers: *MultiEmbedding* simply denotes multiple embedding layers (Bengio et al., 2000) with the same output size, used to obtain a continuous representation for different discrete node attributes. The first hyperparameter refers to the embedding size, while the second describes the way in which embeddings of different features are combined into a single embedding. In particular, *cat* refers to concatenation along the feature dimension. The usage of embedding layers can be exchanged for multiple linear layers, if the input features are already continuous. *PoseConv* is a layer that implements the function described in Equation (3); by respectively naming its three hyperparameters $i$, $h$ and $o$, we can further specify that, in this implementation, $g_s$ and $g_l$ are linear layers with input size $i + 2$ (2 is the size of $p_i$)

Table 4: The FNN-based Encoder $\Psi_\psi$.

| LAYER | H-PARAMS | INPUT | OUTPUT |
|---|---|---|---|
| MultiEmbedding | $8, cat$ | $\boldsymbol{X}$ | $\boldsymbol{O}$ |
| FNNPoseConv | $8, 8$ | $\boldsymbol{O}\|\boldsymbol{Z}, \boldsymbol{P}, \boldsymbol{A}$ | $\boldsymbol{O}$ |
| BatchNorm | | $\boldsymbol{O}$ | $\boldsymbol{O}$ |
| ReLU | | $\boldsymbol{O}$ | $\boldsymbol{O}$ |
| FNNPoseConv | $8, 16$ | $\boldsymbol{O}, \boldsymbol{P}, \boldsymbol{A}$ | $\boldsymbol{O}$ |
| BatchNorm | | $\boldsymbol{O}$ | $\boldsymbol{O}$ |
| ReLU | | $\boldsymbol{O}$ | $\boldsymbol{O}$ |
| FNNPoseConv | $16, 32$ | $\boldsymbol{O}, \boldsymbol{P}, \boldsymbol{A}$ | $\boldsymbol{O}$ |
| Sigmoid | | $\boldsymbol{O}$ | $\boldsymbol{O}$ |

Table 5: The Mask Generator $\mu_\theta$.

| LAYER | H-PARAMS | INPUT | OUTPUT |
|---|---|---|---|
| PoseConv | $3, 8, 8$ | $\boldsymbol{P}\|\boldsymbol{Z}, \boldsymbol{P}, \boldsymbol{A}$ | $\boldsymbol{O}$ |
| BatchNorm | | $\boldsymbol{O}$ | $\boldsymbol{O}$ |
| ReLU | | $\boldsymbol{O}$ | $\boldsymbol{O}$ |
| PoseConv | $8, 32, 32$ | $\boldsymbol{O}, \boldsymbol{P}, \boldsymbol{A}$ | $\boldsymbol{O}$ |
| BatchNorm | | $\boldsymbol{O}$ | $\boldsymbol{O}$ |
| ReLU | | $\boldsymbol{O}$ | $\boldsymbol{O}$ |
| PoseConv | $32, 128, 128$ | $\boldsymbol{O}, \boldsymbol{P}, \boldsymbol{A}$ | $\boldsymbol{O}$ |
| Reshape | $128, [8, 4, 4]$ | $\boldsymbol{O}$ | $\mathbf{O}$ |
| PoseConv2D | $8, 16$ | $\mathbf{O}, \boldsymbol{A}$ | $\mathbf{O}$ |
| ConvBlock2D | $16, 32, 2$ | $\mathbf{O}$ | $\mathbf{O}$ |
| PoseConv2D | $32, 16$ | $\mathbf{O}, \boldsymbol{A}$ | $\mathbf{O}$ |
| PoseConv2D | $16, 8$ | $\mathbf{O}, \boldsymbol{A}$ | $\mathbf{O}$ |
| BatchNorm2D | | $\mathbf{O}$ | $\mathbf{O}$ |
| ReLU | | $\mathbf{O}$ | $\mathbf{O}$ |
| Conv2D | $8, 1$ | $\mathbf{O}$ | $\boldsymbol{O}$ |
| Sigmoid | | $\boldsymbol{O}$ | $\boldsymbol{O}$ |

and output size $h$, while $g_g$ is a linear layer with input size $h$ and output size $o$. Analogously, *PoseConv2D* refers to a layer implementing Equation (5), where $g_o$ is a *ConvBlock2D* with $in = 2 \cdot i$, $out = o$ and $up = 2$, followed by $f(\mathbf{H}) = \sigma(\mathbf{H}) \odot \mathbf{H}$. $g_s$ is a times two upsampling followed by a $1 \times 1$ 2D convolution with 0 padding, input size $i$ and output size $o$ and, finally, $g_g$ is also a $1 \times 1$ 2D convolution with 0 padding and both input and output size set to $o$. In this case $i$ is the first hyperparameter of PoseConv2D, while $o$ is the second. *FNNPoseConv* is the FNN equivalent of PoseConv, computing

$$\boldsymbol{h}_i' = g_s(\boldsymbol{h}_i) + g_{nr}(\boldsymbol{h}_i\|\boldsymbol{p}_i), \tag{13}$$

where, naming the first hyperparameter $i$ and the second $o$, $g_{nr}$ and $g_s$ are both linear layers, the former with input size $i + 2$ and output size $o$ and the latter with input size $i$ and output size $o$. Note that, when using the FNN Conditioner, the PoseConv layers of the discriminator are swapped for equivalent FNNPoseConv ones. Furthermore, note that, unless differently stated, all Conv2D operations use a kernel of size $3 \times 3$ with stride and padding set to 1. All layer weights are normalized through spectral normalization (Miyato et al., 2018), *Upsample* operations are implemented through bilinear interpolation, while *Downsample* ones use 2D average pooling. For completeness, we clarify that the *Sum(H,W)* operation in Table 7, refers to summing the tensor $\mathbf{O}_2 \in \mathbb{R}^{B \times C \times H \times W}$ over the last two dimensions, thus obtaining the matrix $\boldsymbol{O}_2 \in \mathbb{R}^{B \times C}$.

Tables 10 and 11 show, respectively, the architectures of the downstream generator and discriminator, used for the human dataset. In such case the GraPhOSE architecture remained the same, although the

Table 6: The Downstream Generator.

| LAYER | H-PARAMS | INPUT | OUTPUT |
|---|---|---|---|
| ConvBlock2D | $32, 32, 1$ | $\mathbf{L}$ | $\mathbf{O}$ |
| ConvBlock2D | $32, 32, 1$ | $\mathbf{O}$ | $\mathbf{O}$ |
| ConvBlock2D | $32, 16, 1$ | $\mathbf{O}$ | $\mathbf{O}$ |
| ConvBlock2D | $16, 16, 1$ | $\mathbf{O}$ | $\mathbf{O}$ |
| ConvBlock2D | $16, 8, 1$ | $\mathbf{O}$ | $\mathbf{O}$ |
| BatchNorm2D | | $\mathbf{O}$ | $\mathbf{O}$ |
| ReLU | | $\mathbf{O}$ | $\mathbf{O}$ |
| Conv2D | $8, 3$ | $\mathbf{O}$ | $\mathbf{O}$ |
| Tanh | | $\mathbf{O}$ | $\mathbf{O}$ |

Table 7: The Downstream Discriminator.

| LAYER | H-PARAMS | INPUT | OUTPUT |
|---|---|---|---|
| MultiEmbedding | $8, cat$ | $\mathbf{X}$ | $\mathbf{O_1}$ |
| PoseConv | $8, 8, 8$ | $\mathbf{O_1}, \mathbf{P}, \mathbf{A}$ | $\mathbf{O_1}$ |
| BatchNorm | | $\mathbf{O_1}$ | $\mathbf{O_1}$ |
| ReLU | | $\mathbf{O_1}$ | $\mathbf{O_1}$ |
| PoseConv | $8, 16, 16$ | $\mathbf{O_1}, \mathbf{P}, \mathbf{A}$ | $\mathbf{O_1}$ |
| BatchNorm | | $\mathbf{O_1}$ | $\mathbf{O_1}$ |
| ReLU | | $\mathbf{O_1}$ | $\mathbf{O_1}$ |
| PoseConv | $16, 32, 32$ | $\mathbf{O_1}, \mathbf{P}, \mathbf{A}$ | $\mathbf{O_1}$ |
| DConvBlock2D | $3, 8, 2$ | $\mathbf{I}$ | $\mathbf{O_2}$ |
| DConvBlock2D | $8, 16, 2$ | $\mathbf{O_2}$ | $\mathbf{O_2}$ |
| DConvBlock2D | $16, 16, 2$ | $\mathbf{O_2}$ | $\mathbf{O_2}$ |
| DConvBlock2D | $16, 32, 2$ | $\mathbf{O_2}$ | $\mathbf{O_2}$ |
| DConvBlock2D | $32, 32, 2$ | $\mathbf{O_2}$ | $\mathbf{O_2}$ |
| ReLU | | $\mathbf{O_2}$ | $\mathbf{O_2}$ |
| Sum | $H, W$ | $\mathbf{O_2}$ | $\mathbf{O_2}$ |
| Linear | $64, 32$ | $\mathbf{O_1} \| \mathbf{O_2}$ | $\mathbf{O_3}$ |
| ReLU | | $\mathbf{O_3}$ | $\mathbf{O_3}$ |
| Linear | $32, 1$ | $\mathbf{O_3}$ | $\mathbf{O_3}$ |

number of hidden features was increased by a factor of two and $\mu_\theta$ dropped the ConvBlock2D, as the target images had shape $64 \times 64$, hence no further upsampling was required. The main differences are present at the discriminator and generator level, as for this more complex task we borrowed the building blocks of BigGAN (Brock et al., 2019). In particular, DConvBlock2D$^*$ has a slightly different architecture in which the first Conv2D layer outputs $2 \cdot out$ features and is followed by a ReLU and another Conv2D layer that outputs $out$ features. ConvBlock2D$^*$, instead, takes as additional input the downsampled layout mask $\mathbf{L}^*$, which goes through a Conv2D layer that outputs $out$ features; these are concatenated to the output of the Conv2D on the standard input and go through a second Conv2D($2 \cdot out$, $out$) layer, before the skip connection. This serves the purpose of conditioning the generative process at different scales, as, in this case, the downstream generator starts from a $1 \times 4 \times 4$ feature map, $\mathbf{Z}$, which is upsampled at each step, instead of directly using the layout mask and always keeping the same feature map size. For completeness, $\mathbf{Z}$ represents additional latent conditioning, required to account for the background, and is obtained by reshaping the output of a linear layer with output size 16, which takes as input a vector of normally distributed noise of size 8.

Table 12 summarizes the parameter counts for GraPhOSE and the baselines, differentiating between the smaller versions used for the PRO dataset and the larger ones used for Humans.

Table 8: The ConvBlock2D $(in, out, up)$.

| Layer | H-params | Input | Output |
|---|---|---|---|
| BatchNorm | | $\mathbf{H}$ | $\mathbf{O}_1$ |
| ReLU | | $\mathbf{O}_1$ | $\mathbf{O}_1$ |
| Upsample | $up$ | $\mathbf{O}_1$ | $\mathbf{O}_1$ |
| Upsample | $up$ | $\mathbf{H}$ | $\mathbf{O}_2$ |
| Conv2D | $in, out$ | $\mathbf{O}_1$ | $\mathbf{O}_1$ |
| Conv2D | $in, out$ | $\mathbf{O}_2$ | $\mathbf{O}_2$ |
| Sum | | $\mathbf{O}_1, \mathbf{O}_2$ | $\mathbf{O}_1 + \mathbf{O}_2$ |

Table 9: The DConvBlock2D $(in, out, down)$.

| Layer | H-params | Input | Output |
|---|---|---|---|
| ReLU | | $\mathbf{H}$ | $\mathbf{O}_1$ |
| Conv2D | $in, out$ | $\mathbf{O}_1$ | $\mathbf{O}_1$ |
| Downsample | $down$ | $\mathbf{O}_1$ | $\mathbf{O}_1$ |
| Conv2D | $in, out$ | $\mathbf{H}$ | $\mathbf{O}_2$ |
| Downsample | $down$ | $\mathbf{O}_2$ | $\mathbf{O}_2$ |
| Sum | | $O_1, \mathbf{O}_2$ | $\mathbf{O}_1 + \mathbf{O}_2$ |

## A.8 Metrics

We chose Frechet Inception Distance (Heusel et al., 2017) and Inception Score (Salimans et al., 2016) as metrics for our generative models as both are commonly used to asses generation quality. However, only FID is effectively useful in assessing how much the generated samples resemble those coming from the real distribution, as this metric compares the similarity between the distribution of images coming from the generator and those coming from the dataset. Inception Score is limited to evaluation of some properties of the generated images distribution, hence it is less useful as a metric of comparison between models because it can only tell us if the model is capable of generating images that are sufficiently distinct and recognizable, even though they might actually be completely different from those of the real distribution. In other words, IS is only influenced by the capability of the generative model of producing samples that can be distinctively allocated to a class and that, at the same time, are diverse within each class. In our setting, the differences between GraPhose and each baseline are not expected to impact such properties, which are mostly tied to the downstream model, which is always the same for each task. This makes the resulting Inception Score quite uninformative as a way of comparing the various models; nonetheless, it highlights that learning the object's relational masks does not indeed harm the appeal of the generated samples. For the PRO dataset we also reported the Structural Similarity Index Score (SSIM) (Wang et al., 2004). This metric is used as a measure of image reconstruction accuracy, hence it is suitable to evaluate generation quality only in situations where the input to the model uniquely and completely describes the image to be generated. That is the case for the PRO task, as there is a deterministic mapping between the graph and the resulting image. However, that is not the case for the Humans dataset, in which the input graph just describes the person's pose but carries no information regarding the clothing or the background, hence the reason for not employing such metric in that scenario.

## B  Additional results

The following is a collection of additional results, comprising further examples of what already discussed in the main paper, and also images generated by training models with slight architectural changes.

Table 10: The Downstream Generator used for the human dataset.

| LAYER | H-PARAMS | INPUT | OUTPUT |
|---|---|---|---|
| ConvBlock2D* | $1, 512, 2$ | $\mathbf{Z}, \mathbf{L}^*$ | $\mathbf{O}$ |
| ConvBlock2D* | $512, 256, 2$ | $\mathbf{O}, \mathbf{L}^*$ | $\mathbf{O}$ |
| ConvBlock2D* | $256, 128, 2$ | $\mathbf{O}, \mathbf{L}^*$ | $\mathbf{O}$ |
| ConvBlock2D* | $128, 64, 2$ | $\mathbf{O}, \mathbf{L}^*$ | $\mathbf{O}$ |
| Attention | $64$ | $\mathbf{O}$ | $\mathbf{O}$ |
| ConvBlock2D* | $64, 32, 2$ | $\mathbf{O}, \mathbf{L}^*$ | $\mathbf{O}$ |
| BatchNorm2D | | $\mathbf{O}$ | $\mathbf{O}$ |
| ReLU | | $\mathbf{O}$ | $\mathbf{O}$ |
| Conv2D | $32, 3$ | $\mathbf{O}$ | $O$ |
| Tanh | | $O$ | $O$ |

Table 11: The Downstream Discriminator used for the human dataset.

| LAYER | H-PARAMS | INPUT | OUTPUT |
|---|---|---|---|
| MultiEmbedding | $8, cat$ | $\boldsymbol{X}$ | $\boldsymbol{O}_1$ |
| PoseConv | $8, 16, 16$ | $\boldsymbol{O}_1, \boldsymbol{P}, \boldsymbol{A}$ | $\boldsymbol{O}_1$ |
| BatchNorm | | $\boldsymbol{O}_1$ | $\boldsymbol{O}_1$ |
| ReLU | | $\boldsymbol{O}_1$ | $\boldsymbol{O}_1$ |
| PoseConv | $16, 32, 32$ | $\boldsymbol{O}_1, \boldsymbol{P}, \boldsymbol{A}$ | $\boldsymbol{O}_1$ |
| BatchNorm | | $\boldsymbol{O}_1$ | $\boldsymbol{O}_1$ |
| ReLU | | $\boldsymbol{O}_1$ | $\boldsymbol{O}_1$ |
| PoseConv | $32, 64, 64$ | $\boldsymbol{O}_1, \boldsymbol{P}, \boldsymbol{A}$ | $\boldsymbol{O}_1$ |
| DConvBlock2D* | $3, 32, 2$ | $\mathbf{I}$ | $\mathbf{O}_2$ |
| Attention | $32$ | $\mathbf{O}_2$ | $\mathbf{O}_2$ |
| DConvBlock2D* | $32, 64, 2$ | $\mathbf{O}_2$ | $\mathbf{O}_2$ |
| DConvBlock2D* | $64, 128, 2$ | $\mathbf{O}_2$ | $\mathbf{O}_2$ |
| DConvBlock2D* | $128, 256, 2$ | $\mathbf{O}_2$ | $\mathbf{O}_2$ |
| DConvBlock2D* | $256, 512, 2$ | $\mathbf{O}_2$ | $\mathbf{O}_2$ |
| ReLU | | $\mathbf{O}_2$ | $\mathbf{O}_2$ |
| Sum | $H, W$ | $\mathbf{O}_2$ | $\mathbf{O}_2$ |
| Linear | $576, 512$ | $\boldsymbol{O}_1 \| \boldsymbol{O}_2$ | $\boldsymbol{O}_3$ |
| ReLU | | $\boldsymbol{O}_3$ | $\boldsymbol{O}_3$ |
| Linear | $512, 1$ | $\boldsymbol{O}_3$ | $\boldsymbol{O}_3$ |

## B.1 Missing attributes

As discussed, our formulation allows for sub-conditioning the pose graph, i.e., having nodes without any semantic attribute assigned, in order to let the generative process decide how to treat them. Figure 11 shows some examples of this behavior, in which white nodes represent nodes without any semantic attribute. We can see that the generation is mostly consistent when the attributes can be recovered from neighbouring nodes. However, in cases in which the information needs to be reconstructed through multiple hops (e.g., the scissors handle) the model uses the nearest color instead. This might be the result of biases in the data. Mind that we use the term *reconstruction* loosely here, as the model is trained with a generative objective, hence we might expect different results if we train it on a strict reconstruction task. Figure 12, instead, shows what happens when we have a growing chain of missing attributes. We can see that semantics are coherently filled in, by GraPhOSE, up to a certain number of subsequent masked nodes. This is, in part, due to architectural constraints, as the number of hops information can traverse depends on the number of message passing steps. On the other hand, the usage of graph convolution operators that try to prevent

Table 12: GraPhOSE and baselines approximate parameter count, for the two tasks, divided between conditioning, downstream generator and discriminator.

| Models | PRO | | | Humans | | |
|---|---|---|---|---|---|---|
| | Conditioning | Generator | Discriminator | Conditioning | Generator | Discriminator |
| FNN Conditioning | 2K | 27K | 23K | 6K | 8.6M | 5.2M |
| GNN Conditioning | 5K | 27K | 26K | 17K | 8.6M | 5.2M |
| **GraPhOSE** | 63K | 27K | 26K | 2.4M | 8.6M | 5.2M |

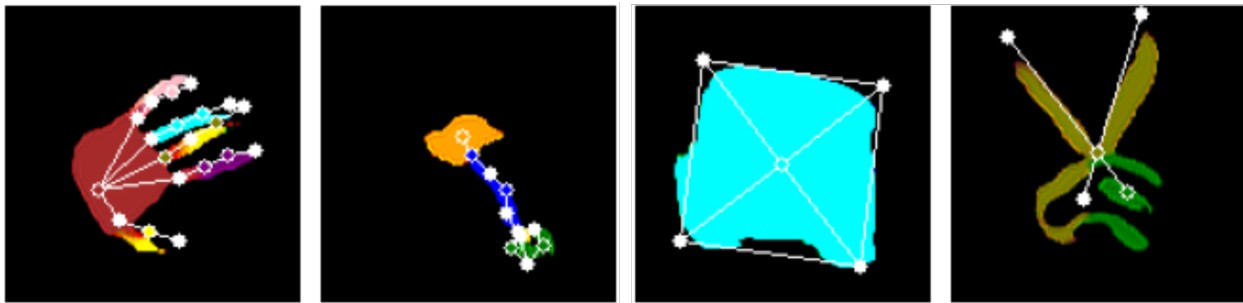

Figure 11: Sample images generated from GraPhOSE in presence of missing semantic attributes: from left to right, Hand, Robotic Arm, Filled Polygon and Scissors. White nodes have their semantic attributes masked, while others are colored according to their color attribute.

over-smoothing might also play a role in this. However, notice we did not train our model to specifically address high chances of partial conditioning, i.e., missing attributes; different results can be expected if we purposefully train it on datasets with such characteristics. Even though, in such experiments, the color relationships between object components are rather simple, we think these results, tied with the architectural constraints involved, can be a promising basis for further, and more specifically aimed, experimentation.

## B.2 Mask adaptation

Figure 13 shows additional examples of how the pre-learned masks consistently adapt to the downstream task during the end-to-end training, while figure 14 shows how directly training on the downstream task leads to masks that are not visually meaningful. This is particularly severe for the case of human poses, where the dataset is heavily biased towards a few particularly common ones.

Figure 15 shows further examples of how node-mask locality is preserved after downstream training, and adapted to the specifics of the objects in the task. We can notice that the Pie (e) is the only object for which this locality seems to be lost, however it is not surprising as its graph representation diameter is only 2, with just 3 nodes, hence it is harder to avoid over-smoothing. For graphs with larger diameter and/or number of nodes, we see that locality is retained.

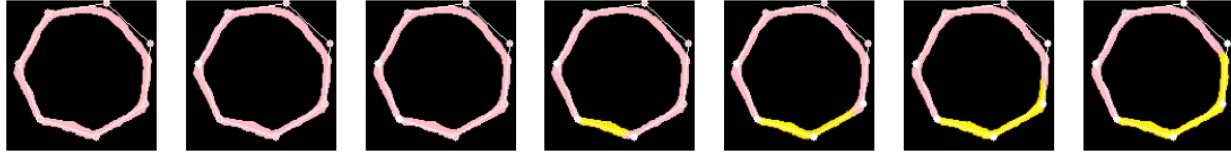

Figure 12: Example of how GraPhOSE behaves, with a graph with $n$ nodes, in presence of missing attributes, as the chain of subsequent ones increases in length from 0 (left) to $n - 1$ (right). White nodes represent missing attributes.

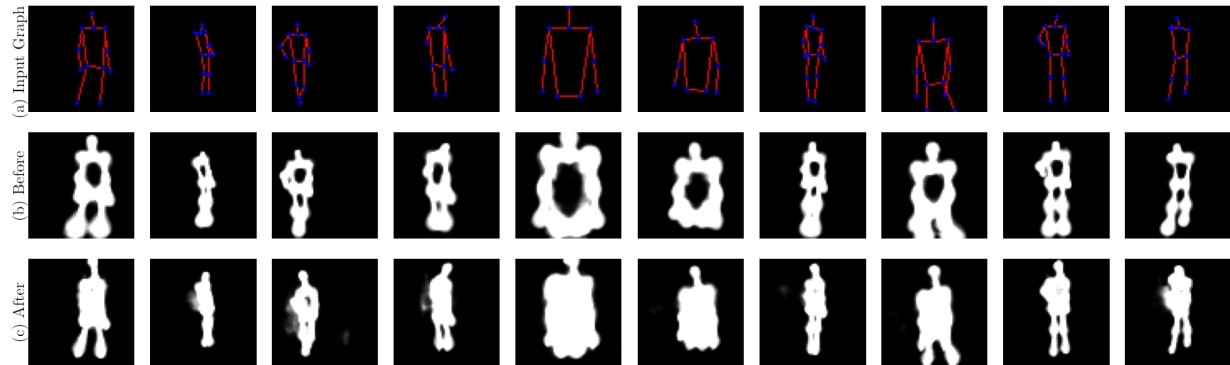

Figure 13: Random sample of humans masks generated by our model after pre-training (b) and after subsequent training on the downstream task (c). The top row (a) shows the corresponding input graph.

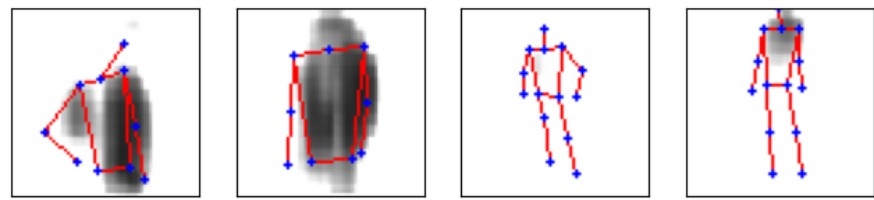

Figure 14: Samples of humans masks generated by GraPhOSE's mask generator after end-to-end training without surrogate mask pre-training. The input graph is superimposed in blue (nodes) and red (edges).

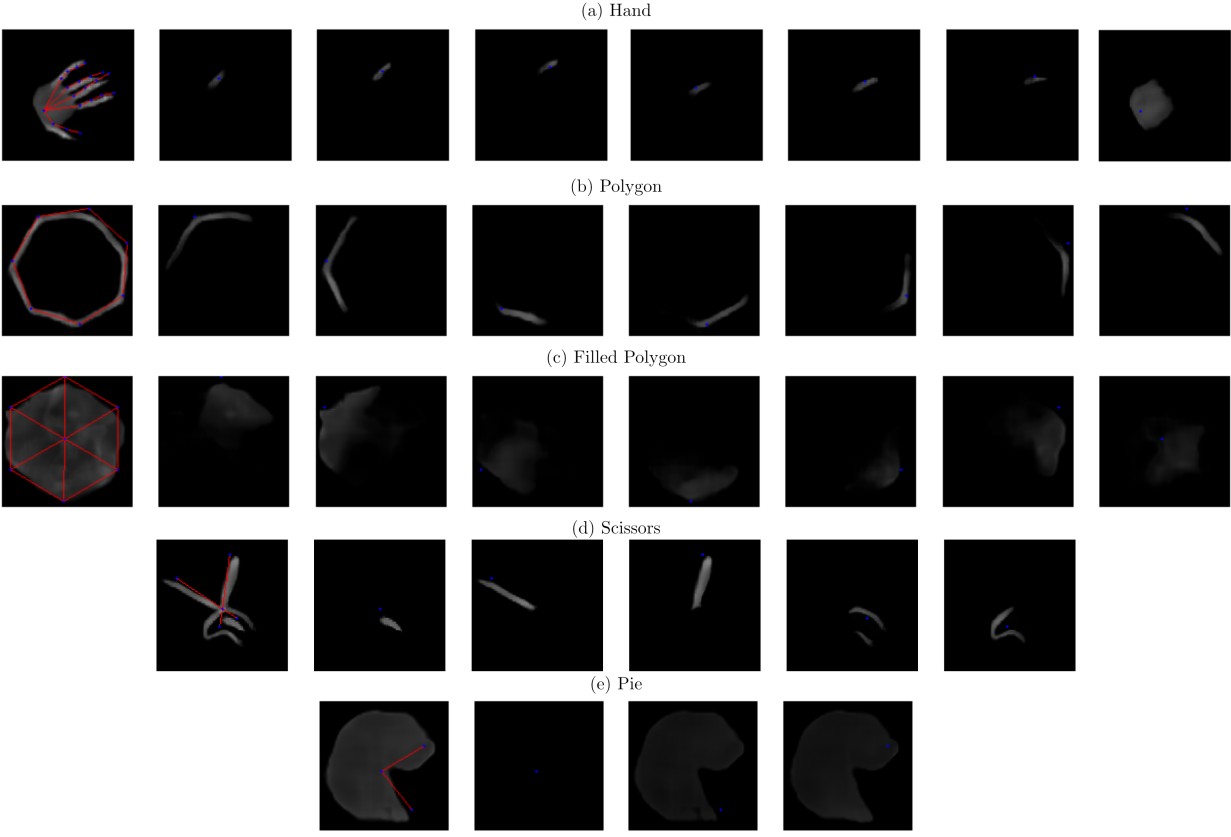

Figure 15: Samples of individual node masks generated from GraPhOSE. The first column is the cumulative mask for the whole input graph (superimposed with nodes in blue and edges in red). Individual node masks have the reference node highlighted in blue. Hand masks (a) are only a sub-sample for ease of visualization.

