# OpenReview forum: "Object-Centric Relational Representations for Image Generation"
_TMLR — Accepted by TMLR_

### Review · Reviewer_CsdN · 2024-03-17

**Summary Of Contributions:**

This paper introduces a novel method for conditioning image generation using object-centric relational representations. The paper addresses the limitation of existing approaches by proposing a unified method for representing both structural and semantic conditioning at various granularity levels in image generation tasks.

**Audience:**

Yes

**Claims And Evidence:**

Yes

**Requested Changes:**

Can author exposure more details of the Test Set? For example:

1. The distribution of the dataset: The paper should provide information on the size of the test dataset, including the number of images and the distribution of objects and classes within these images. This information helps assess the diversity and representativeness of the dataset.

2. Table 2 and Object Classes: If Table 2 lists all the objects and classes used in the experiment, it is essential to confirm whether this coverage is sufficient for constructing a comprehensive test set. An ideal test set should encompass a wide range of objects and classes to evaluate the generative model's ability to handle diverse scenarios.

3. Generalizability: It's crucial to assess whether the test set generalizes well to measure the quality of the generative model accurately. The test set should cover a variety of object configurations, backgrounds, lighting conditions, and other factors to evaluate the model's robustness and generalization capabilities effectively.

Can this paper leverage widely used benchmarks or tasks to enhance the validation and applicability of the proposed model?

**Strengths And Weaknesses:**

Strength:

Clear Methods and Unified Framework: The paper's strength lies in its clear exposition of methods and the proposal of a unified framework for conditioning image generation. This clarity facilitates understanding and implementation, enabling researchers to easily adopt and extend the proposed approach in their work.

Weakness:

Lack of Clarity on Dataset Design and Generation: The paper could benefit from providing more detailed information on how the test datasets were designed and generated. Additionally, the lack of support from other open datasets may limit the demonstration of the method's efficacy across diverse domains and data distributions. Incorporating additional datasets or providing insights into dataset selection and generation processes could strengthen the paper's empirical validation and broaden its applicability.

---

> ### Author Response · Authors · 2024-03-23
> **Answers to reviewer's comments**
>
> We thank the reviewer for the useful comments. Regarding the requested changes, which align with the highlighted weaknesses, we address them in the following.
>
> >The distribution of the dataset: The paper should provide information on the size of the test dataset, including the number of images and the distribution of objects and classes within these images. This information helps assess the diversity and representativeness of the dataset.
>
> The models were tested on ~3000 samples for the Humans task and 1000 samples for the PRO (synthetic) task. Within the Humans task, the only object type is "Person", while there are 14 node classes (left and right shoulders, elbows, wrists, ankles, knees, hips plus base of the neck and top of the head). For the PRO task, Table 2 covers the 7 object types, together with the 13 node classes. For the Humans dataset samples were uniformly drawn from the different benchmarks, to retain the same distribution observed during training. For the PRO dataset, the number of objects in an image was uniformly sampled between 1 and 4 (to avoid over cluttering the small images), and object types were also uniformly sampled from the 7 classes. The object arrangements were uniformly sampled over ranges of plausible values, to provide diversity without the creation of impossible structures (e.g., a thumb cannot form a >90-degree angle  w.r.t. the hand's palm). More details are reported in Section A.5. We will integrate the information that was not already specified in the main paper.
>
> >Table 2 and Object Classes: If Table 2 lists all the objects and classes used in the experiment, it is essential to confirm whether this coverage is sufficient for constructing a comprehensive test set. An ideal test set should encompass a wide range of objects and classes to evaluate the generative model's ability to handle diverse scenarios.
>
> Regarding image diversity, the Humans dataset was obtained by blending three different benchmarks (MPII Human Pose, Deep Fashion and Market 1501), which are widely used in person generation tasks. For the PRO dataset, we considered a set of objects that could encompass a varying level of complexity, in terms of number of parts/nodes, size, and arrangement. Moreover, we provided semantic variability by randomizing colors for the object parts. Furthermore, note that depending on the objects there are different ways in which input pose graphs should be filled in (e.g., consider a robotic arm and a semi-circle). We thus consider the proposed test sets to cover a sufficiently wide range of scenarios.
>
> >Generalizability: It's crucial to assess whether the test set generalizes well to measure the quality of the generative model accurately. The test set should cover a variety of object configurations, backgrounds, lighting conditions, and other factors to evaluate the model's robustness and generalization capabilities effectively.
>
> As stated above, we designed our synthetic task to provide reasonable variability in the resulting images. However, since our main contribution is in conditioning on the objects in the image, there are no background or lightning changes in the PRO dataset. The overall visual “style” of the image could be handled by a standard vector representation fed to the downstream model and, as such, was out of the scope of our study. Note that different backgrounds and lighting conditions are present in the Humans task, but not specified in the conditioning. Finally, in Figure 7, we show how our model naturally learns disentangled masks for each part of the object, which is an indicator of the modularity of the learned components.
>
> >Can this paper leverage widely used benchmarks or tasks to enhance the validation and applicability of the proposed model?
>
> The Humans task, as explained in Section 5.3, consists of images from 3 datasets (MPII Human Pose, Deep Fashion and Market1501), widely adopted as benchmarks for person image generation. It would be possible to train the model on other datasets that provide images together with graph annotations of different objects. However, there are no well-established datasets that we are aware of. This is the reason that brought us to design a synthetic task. We hope that our work can motivate the collection of such datasets in the future. Extending the synthetic dataset to more pose-guided multi-object image generation scenarios is a direction for future work.
>
> We will clarify all the above in the document. Please let us know if we successfully addressed all of your concerns.

---

### Review · Reviewer_bkj9 · 2024-03-19

**Summary Of Contributions:**

This paper proposes a new method for conditioning image generation using object-centric relational representations. By leveraging attributed graphs to represent structure and semantic information, it allows for effective conditioning of object generation in images. Implemented through a neural network with 2D and graph convolutional operators, empirical results demonstrate its effectiveness compared to baselines on a synthetic dataset.

**Audience:**

Yes

**Claims And Evidence:**

No

**Requested Changes:**

Please see the weakness part for the required changes.

**Strengths And Weaknesses:**

Pros:
1)   This is the first work exploring conditional image generation using object-centric relational representations.
2)   They introduce a new benchmark for image generation, comprising a synthetic dataset pairing images with their relational representations.

Cons:
1)   The proposed method’s advantages in flexibility, generality, and effectiveness compared to existing methods remain uncertain due to the lack of comparative experiments with Scene Graph Conditioned Image Generation and Pose Conditioned Image Generation methods. Although the existing models have limited object diversity and often require diverse inputs, conducting experiments with consistent design is crucial for validating the claim.
2)   Figure 8 demonstrates the consistency of generated images across different node positions. However, there is a lack of experimental validation regarding the variation of node semantic information and its impact on the generated results.
3)  Could the author offer a more detailed description of the dataset, such as its size, and how the training and testing data are partitioned?
4)  I suggest the authors following [1] to perform a more fair and more comprehensive evaluation. Specifically, more metrics on more datasets should be used, such as SSIM and user study etc.
5)  No explanation about the meaning of A in Equation 2.

[1] “Pose Guided Person Image Generation”, NeurIPS 2017.

---

> ### Author Response · Authors · 2024-03-23
> **Answers to reviewer's comments 1/2**
>
> We thank the reviewer for the useful comments. Regarding the highlighted weaknesses, we address them in the following.
>
> >The proposed method’s advantages in flexibility, generality, and effectiveness compared to existing methods remain uncertain due to the lack of comparative experiments with Scene Graph Conditioned Image Generation and Pose Conditioned Image Generation methods.
>
> Advantages in flexibility and generality over other methods are inherent to the fact that other approaches cannot condition generation on a variable number of objects together with structural and semantic information. Regarding effectiveness (i.e., image quality), we show better results against relevant baselines that can tackle such a task (considering the same generative backbone to ensure a fair assessment). We argue that a comparison with other Pose Conditioned approaches (possible only considering a fixed number of keypoints)  would not add a lot to the discussion since obtaining state-of-the-art image quality in the generate samples is not the objective here. Rather, we are proposing a new framework that covers a broader type of structured conditioning. Note that we already include baselines that cover other non-graph-based approaches for providing a similar conditioning and results support the adoption of our approach.
>
> >Although the existing models have limited object diversity and often require diverse inputs, conducting experiments with consistent design is crucial for validating the claim.
>
> We briefly justified the lack of comparison with Scene Graph Conditioned Image Generation and Pose Conditioned Image Generation in Section 5.2. We explain our reasoning here in more detail:
>
> * Scene Graphs based methods, as mentioned in Section 4, do not condition the generation on any geometrical property. They employ graphs obtained by parsing natural language sentences and solve a different task, closer to text-to-image generation. Thus, any comparison would be merely in terms of image quality, which is not the focus of our work, in fact we employ a simple GAN as downstream model. Such comparison would yield little information, and it would be akin to comparing with a text-based image generation model.
> * Similarly, Pose Conditioned methods (like [“Pose Guided Person Image Generation”, NeurIPS 2017]) solve an image-to-image task, in which the pose of the person in the input image is translated to a target pose. Thus, they tackle a very different task. Moreover, such models cannot be easily modified to handle a variable number of objects, hence a comparison could be done only for the Humans task and require an entirely different conditioning pipeline. Such comparison could be only in terms of image quality, as there is no principled way to quantitatively evaluate pose alignment, hence, it would be of little interest as the downstream image generator would be the same.
>
> We understand that the related works Section might mislead the reader into thinking that the mentioned methods can be seen as a baseline for our task, while they tackle very different scenarios, which do not align with the setting we are proposing. We mentioned them as related tasks as there are similarities, but a direct comparison either cannot be drawn or would not be meaningful, as we are not proposing an extension of such methods, but rather a methodology targeting a different problem. We will expand the discussion on these aspects in the paper, to make this point clearer.
>
> >Figure 8 demonstrates the consistency of generated images across different node positions. However, there is a lack of experimental validation regarding the variation of node semantic information and its impact on the generated results.
>
> We analyzed this aspect in the PRO dataset where we show that our model can generate images that reflect the semantic information in the input graph (i.e., the colors of the different object's parts and different object types). We will add an additional figure to highlight this aspect, showing the result of the generation after incremental modifications to the attributes of the generated objects. For the Humans dataset the semantic information available consists of the node types (e.g., shoulder, knee, …), but these follow a fixed pattern, hence, if the model is provided with out-of-distribution inputs (e.g., replacing a wrist with an ankle) it would most likely filter out the noisy input and leverage the relational information to produce images that resemble a  prototypical human.

---

> ### Author Response · Authors · 2024-03-23
> **Answers to reviewer's comments 2/2**
>
> >Could the author offer a more detailed description of the dataset, such as its size, and how the training and testing data are partitioned?
>
> The Humans dataset consists of ~300000 images, coming from MPII Human Pose, Deep Fashion and Market 1501, while the PRO dataset consists of 100000 images. Out of these datasets, ~3000 and 1000 images, respectively, were used for validation and an analogous size for testing. The train, validation and test sets were partitioned at random, however, for the Humans datasets, the splits preserve the image distribution among the 3 original benchmarks. Details about the datasets are present in Section 5.3, while specifics about the synthetic data generation can be found in Section A.5. We will integrate the missing information about datasets size and splitting strategy.
>
> >I suggest the authors following [“Pose Guided Person Image Generation”, NeurIPS 2017] to perform a more fair and more comprehensive evaluation. Specifically, more metrics [...] such as SSIM and user study etc.
>
> Regarding the metrics used in the suggested study, we already employ the Inception Score and additionally use the Fréchet Inception Distance. The SSIM metric is not particularly suited for our scenario, as it is mostly a measure of image reconstruction, hence its usage in Pose Guided Image Generation, which is an image-to-image task. We can add SSIM for the PRO dataset, as the graph structure, in such a scenario, fully describes the image to be generated; however, in the Humans setting, where the input graph could represent many visually different target images, there is no reason to expect good SSIM scores. Conversely, regarding the masked version of IS and FID, we can compute them for the Humans task only, as it would not be meaningful in PRO images which have a black background (we will add this in the revision of the paper).
>
> >[...] more datasets should be used
>
> Regarding the referenced work, our Humans dataset is already a combination of the two datasets used in such work, plus, additionally, the MPII Human Pose dataset, as described in Section 5.3.  The adopted benchmarks show that our approach is suitable to tackle the proposed generative task in both a multi-object synthetic case and a single-object real-world one. To the best of our knowledge, there is no well-established, annotated, real-world multi-object image generation dataset. That is the reason that brought us to design a synthetic one. We hope that our work can motivate the collection of suitably annotated datasets in the future, while the extension of the synthetic dataset to more diverse scenarios is a direction for future work.
>
> >No explanation about the meaning of A in Equation 2.
>
> $A$ is the graph's adjacency matrix, as stated in Section 2. We will make this clear in the paper.
>
> Please let us know if we successfully addressed all of your concerns.

---

### Review · Reviewer_GbKP · 2024-04-25

**Summary Of Contributions:**

The paper introduces a novel method for conditioning image generation using object-centric relational representations. The methodology involves conditioning the generation of objects in an image on an attributed graph that represents their structure and semantic information. The approach uses a neural network to generate a 2D, multi-channel layout mask of the objects, which serves as a soft inductive bias in the downstream generative task. The proposed method outperforms some baselines on a new PRO benchmark consisting of a synthetic dataset of images paired with their relational representation.

**Audience:**

Yes

**Broader Impact Concerns:**

I have no concern about the broader impacts and the ethical implications.

**Claims And Evidence:**

Yes

**Requested Changes:**

See Weaknesses.

**Strengths And Weaknesses:**

Strengths:
1. The method looks interesting.
2. The generated images can be flexibly controlled by graphs.

Weaknesses:
1. My major concern lies in the quality of the generated images. According to Figure 5, the generated image seems blurred. In fact, some relevant methods from some years ago [*1, *2] may already outperform this paper in terms of generation quality.
2. Each image in the training set needs to be annotated with a graph, which not only increases the cost, but also hinders the proposed method from being applied to more general image generation tasks.

[*1] CVPR'18, Image Generation from Scene Graphs.
[*2] ECCV'16, Attribute2Image: Conditional Image Generation
from Visual Attributes

---

> ### Author Response · Authors · 2024-04-27
> **Answers to reviewer's comments**
>
> We thank the reviewer for their comments. Regarding the highlighted weaknesses, we address them in the following:
>
> >My major concern lies in the quality of the generated images. According to Figure 5, the generated image seems blurred. In fact, some relevant methods from some years ago may already outperform this paper in terms of generation quality.
>
> We would like to clarify that the focus of our work is on the conditioning aspect of the generative model, i.e., flexibly generating an hidden representation that can be used by the downstream generative model to produce an image that properly reflects the graph structure and attributes. In fact we employ a rather simple CNN-based GAN and 64x64 pixels images, as image quality was not the subject of our study. We will clarify this aspect in the document.
>
> >Each image in the training set needs to be annotated with a graph, which not only increases the cost, but also hinders the proposed method from being applied to more general image generation tasks.
>
> There might be a misundarstending regarding the annotations required by our method. The annotation effort is equivalent to the one required by less flexible approaches that rely on just the keypoints, as the graph connectivity, given the set of keypoints, can be extracted in an automated fashion, without further human intervention.
>
> We would like to thank the reviewer again and ask them to let us know if we successfully addressed their concerns.

---

### Author Response · Authors · 2024-05-03

Dear reviewers,
let us know if we addressed all of your concerns.
We are available for further discussion during this interaction period, and eager to follow up with more details/clarifications.
We would like to thank you again for your comments.

---

### Decision · Action_Editor_x1KK · 2024-06-20

**Recommendation:** Accept with minor revision

**Comment:**

This paper has two contributions:

1) a methodological framework for conditioning images on object-centric relational representations.
2) a new synthetic benchmark dataset with images and conditioning representations.

The reviewers appreciated the contributions of the paper, but also had some reservations especially regarding image quality and comparison to prior work. The authors addressed these questions to some degree. In the final revised version the authors are encouraged to revise the paper reflecting these comments. This will make the paper stronger.

**Audience:**

Yes.

**Claims And Evidence:**

Yes.

---

> ### Author Response · Authors · 2024-07-01
>
> Dear Dr Winther,
> we thank you and the reviewers for the help in improving our work.
>
> Concurrently to this message we are submitting the camera ready version we revised to reflect yours and the reviewers' comments.
> In particular:
> * We specified the meaning of A in Eq.2
> * We explained more clearly the differences between the related works and our approach in Section 4 and 5.2, in order to make it clear why a comparison with such approaches would not be meaningful
> * We added missing details regarding the datasets in Section 5.3, A.5 and A.6
> * We added the SSIM score in Table 1 and discussed such results in Section 6. We also added some details regarding SSIM in Section A.8
> * We refactored Section 6.2 and added an experiment showing the sensitivity of the model to changes in the style attributes when the structure itself is kept fixed.
> * In section A.1, for reproducibility, we added a link to the code that will be made available upon publication.
>
> Let us know if these changes reflect what was expected.